



# Comparison of the Arctic upper-air temperatures from radiosonde and radio occultation observations

Liang Chang[1,2], Lixin Guo[1,2], Guiping Feng[1,2], Xuerui Wu[3,4] and Guoping Gao[1,2]

[1]College of Marine Sciences, Shanghai Ocean University, Shanghai, 201306, China
[2]Collaborative Innovation Center for Distant-water Fisheries, Shanghai, 201306, China
[3]Shanghai Astronomical Observatory, Chinese Academy of Sciences, Shanghai, 200030, China
[4]Key Laboratory of Planetary Sciences, Shanghai Astronomical Observatory, Chinese Academy of Sciences, Shanghai, 200030, China

*Correspondence to*: Guoping Gao (gpgao@shou.edu.cn)

**Abstract.** Air temperature is one of the most important parameters used for monitoring Arctic climate change. The Constellation Observing System for Meteorology, Ionosphere, and Climate and Formosa Satellite mission 3 (COSMIC/FORMOSAT-3) radio occultation (RO) "wet" temperature product (i.e., "wetPrf") was introduced to analyze the Arctic air temperature profiles at 925–200 hPa in 2007–2012. The "wet" temperatures were further compared with radiosonde (RS) observations. Results from the spatially and temporally synchronized RS and COSMIC observations showed that their temperatures were matched well with each other, especially at 400 hPa. Comparisons of seasonal temperatures and anomalies from COSMIC and homogenized RS observations suggested the limited number of COSMIC observations during the spatial matchup may be insufficient to describe the small-scale spatial structure of temperature variations. Furthermore, comparisons of seasonal temperature anomalies from RS and 5 × 5 degree gridded COSMIC observations at 400 hPa during the sea ice minimum (SIM) of 2007 and 2012 were also made. Results revealed that the widely covered COSMIC observations can provide more details than RS observations in describing the Arctic temperature variations. Therefore, despite COSMIC observations being unsuitable to describe the Arctic temperatures in the lowest level, they provide a complementary data source to study the Arctic upper-air temperature variations and related climate change.

## 1 Introduction

Arctic air temperature variations play an important role in numerous processes in the Arctic region by controlling the transfer of mass and moisture fluxes through the lower troposphere. It is therefore essential that Arctic air temperature is accurately monitored. General circulation models (GCMs) are a feasible way to predict air temperature at low latitudes, but they appear to under-predict temperatures over the Arctic (Melles et al., 2012; Ballantyne et al., 2013). The operational radiosonde (RS) is a traditional tool to detect temperature variations in the troposphere and lower stratosphere (e.g., Bohlinger et al., 2014), while it is mainly available over land areas and suffers from poor temporal resolution. Space-borne monitoring is an effective and increasingly important way to obtain temperature profiles with improving accuracy over both land and ocean,



which has been demonstrated in the polar regions with the Moderate Resolution Imaging Spectroradiometer (MODIS) (e.g., Liu et al., 2003), the High Resolution Infrared Radiation Sounder (HIRS) (e.g., Liu et al., 2006) and the Atmospheric Infrared Sounder (AIRS) data (e.g., Devasthale et al., 2010; 2013), etc. However, these infrared-based sensors are typically sensitive to the presence of clouds, which may limit their applications in the polar regions due to the low frequency of cloud-
free conditions in the polar regions (Liu et al., 2012; Chan et al., 2013).

    The Global Navigation Satellite System (GNSS) Radio Occultation (RO) is the first space-borne remote sensing technique that can provide high vertical resolution (less than 1km) all-weather refractivity profiles, which depend on pressure, temperature and humidity (Kuo et al., 2000; Yunck et al., 2000). The RO technique was based on the path of a radio signal propagating between a GNSS satellite (e.g., Global Positioning System (GPS)) and a receiver placed on a low earth orbit
(LEO) satellite bent by the atmosphere (Kursinski et al., 1997). The bending angles of the RO signal are derived from the propagation time, which can be precisely measured with atomic clocks. The retrieved bending angle profiles are used to derive profiles of refractivity, and subsequently meteorological parameters such as pressure, temperature and humidity (Kuo et al., 2000). It was not possible for the RO technique to separate water vapor and temperature effectively in the lower troposphere, caused by the ambiguity between temperature and water vapor in refractivity. As a result, the "dry" temperature
can be derived in the stratosphere and upper troposphere where the water vapor partial pressure can be neglected, while the "wet" temperature can be estimated with moisture information included. Considering the RO technique has the advantages of global coverage, high accuracy, long-term stability and self-calibration (Kursinski et al., 1997; Wickert et al., 2001; Hajj et al., 2004), it may offer us a unique opportunity to monitor the state of Arctic air temperature.

    In this paper, the "wet" temperature profiles derived from the Constellation Observing System for Meteorology,
Ionosphere, and Climate (COSMIC)/Formosa Satellite Mission 3 (hereafter COSMIC) RO observations were used to estimate the Arctic temperatures and anomalies from 2007 to 2012, which were further compared with the results from RS observations. This paper was organized as follows. In Section 2, the RS and COSMIC derived temperature profiles from 925 to 200 hPa over the Arctic were assessed via comparison with the Global Climate Observing System (GCOS) Reference Upper-Air Network (GRUAN) products. Additionally, the temperature profiles from RS and COSMIC observations were
compared to understand their performance in describing the Arctic atmospheric temperature. Furthermore, comparisons of observations from COSMIC and different types of RS were also discussed. In order to understand the performance of RS and COSMIC observations in monitoring Arctic climate change, Section 3 compared the seasonal mean temperatures and anomalies from spatially matched RS and COSMIC observations, and analyzed their differences between RS and $5 \times 5$ degree gridded COSMIC observations. Section 4 was devoted to discussing the temperature anomaly differences from RS
and $5 \times 5$ degree gridded COSMIC observations during the record minimum sea ice extents of 2007 and 2012, and comparing the performances of RS and COSMIC observations in revealing the temperature variations during the sea ice minimum (SIM) events. Finally, the conclusions of the present analysis were summarized in Section 5.



## 2 Analysis of cosmic derived Arctic temperature profiles

### 2.1 RO data

The COSMIC, launched in April 2006, is a joint US/Taiwan GPS RO mission consisting of six identical micro-satellites. The COSMIC mission can provide in near real time the vertical profiles of bending angles, refractivity, temperature, pressure,

and water vapor in the neutral atmosphere and electron density in the ionosphere with global coverage for atmospheric and ionospheric research, as well as for improving global weather forecasts and climate change related studies. A distinctive feature of the COSMIC mission, compared to previous RO missions, is tracking both setting and rising neutral atmospheric occultations in the lower troposphere in an open-loop (OL) mode (Schreiner et al., 2007). The OL tracking is very important for the GPS receiver to correctly process and record tropospheric RO signals, and to avoid the larger uncertainty in the

retrieved refractivity below 5 km, particularly over the tropics.

In the present study, the COSMIC post-processed level-2 one-dimensional variational analysis (1DVAR) "wetPrf" product (Das and Pan, 2014) during the period from 13 July 2006 to 31 December 2013 was collected from the COSMIC Data Analysis and Archival Center (CDAAC) for further analysis. The European Centre for Medium-Range Weather Forecasts (ECMWF) analysis data were used for background in the 1DVAR process during the "wetPrf" product generation, which

separated the pressure, temperature and moisture contributions from refractivity. The "wetPrf" product is atmospheric occultation profiles with a priori information about the water vapor in the lower levels of the troposphere included (Das and Pan, 2014). Unlike the "dry" temperatures (i.e., "atmPrf" product), which are known to be of great quality in the upper troposphere and lower stratosphere (e.g., Kursinski et al., 1997; Hajj et al., 2004; Sun et al., 2013; Kuo et al., 2004), the performance of the "wet" temperatures (i.e., "wetPrf" product) need to be further investigated. Therefore, the "wet"

temperature profiles, rather than the "dry" temperature profiles, were used in this study to compare with the RS temperature measurements at 925–200 hPa over the Arctic. The post-processed version of "wetPrf" data was 2010.2640, and the altitude range is 0–40km at 100m vertical resolution. Each "wetPrf" file contains a profile of "wet" temperature, altitude, pressure, latitude, longitude, and azimuth angle, while the file header contains the latitude and longitude of the perigee point at occultation and the RO start and stop time. Although the COSMIC data are available at CDAAC starting on 21 April 2006,

the data before 13 July 2006 were excluded in this study because they were collected with receiver firmware version older than 4.2.1, which suffered from many different issues not supported by CDAAC.

The spatial distribution of the COSMIC observations from 13 July 2006 to 31 December 2013 in the Arctic Region (65 °N–90 °N) were shown on 5 × 5 degree grids (i.e., 73 columns by 6 rows) in Fig. 1. The number of successful COSMIC occultation events is much less than listed in the occultation table at CDAAC because 1) the limitations in the number of

GPS channels available on the COSMIC satellites, 2) some COSMIC events are not tracked due to power limitations or altitude problems on the satellite, 3) a few occultations might not be downlinked correctly due to satellite transmitter or ground station problems, 4) some of the tracked and received occultations on the ground cannot be processed due to missing auxiliary data, noisy L2 frequency data, or other data problems. As a result, a total of 309102 valid profiles were recorded





from 13 July 2006 to 31 December 2013, and a maximum of 1676 profiles in single grid was observed. As illustrated in Fig. 1, the number of COSMIC profiles decreases as the latitude increases in the Arctic, and it was marked with very low coverage over the areas near the North Pole (80 °N – 90 °N). One of the most probable reasons for the diminishing number of COSMIC profiles might be the decreasing area of the grid cells closer to the North Pole.

*[Insert Fig. 1 here]*

**2.2 RS observations**

Despite the low spatial and poor temporal resolution for the RS observations, they are a key data set in operational weather forecasting and upper-air climate research (Sun et al., 2010). Furthermore, the high-quality RS observations are also very valuable for calibration and validation of satellite temperature (e.g., Sun et al., 2010) and water vapor retrievals (e.g., Chang

et al., 2015). The Integrated Global Radiosonde Archive (IGRA) is a quality-controlled compilation of RS observations from the global network of more than 1500 stations and includes the observed temperature, geopotential height, humidity, wind direction, and wind speed at standard (mandatory) pressure levels and significant levels (Durre et a., 2006). In this study, the RS temperature profiles from 925 to 200 hPa extracted from IGRA were used to compare with the COSMIC temperature profiles. Given that the values of geopotential height or temperature at some pressure levels were not recorded from time to

time, the RS data levels in the absence of either geopotential height or temperature are removed during processing.

**2.3 Assessment of COSMIC derived Arctic temperature profiles**

In this study, the RS and COSMIC derived temperature profiles were compared over the Arctic. The horizontal distance was limited to within 100 km and the time window was 2.5 h during the matchup process between the RS and COSMIC observations. In addition, given that a lowest altitude of only 0.1 km can be reached for the COSMIC temperature profiles,

the temperature comparisons were made at standard pressure levels of the RS profiles from 925 to 200 hPa only (i.e., 925, 850, 700, 600, 500, 400, 300, 250 and 200 hPa, respectively), excluding the 1000 hPa and surface levels. Moreover, considering the vertical resolution of COSMIC profiles is much higher than the RS standard pressure levels, the COSMIC temperature profiles were interpolated on the standard pressure levels of the RS profiles via (Wang et al., 2013)

$$\alpha = \frac{\ln P - \ln P_2}{\ln P_1 - \ln P_2}, \quad \beta = \frac{\ln P_1 - \ln P}{\ln P_1 - \ln P_2} \tag{1}$$

$$T = \alpha \cdot T_1 + \beta \cdot T_2 \tag{2}$$

where $P$ and $T$ are the pressure and temperature at the standard pressure level, while $P_i$ and $T_i$ ($i$ = 1, 2) represent the pressure and temperature from COSMIC "wetPrf", respectively.

Fig. 2 showed the scatterplots of temperatures between COSMIC and all available Arctic RS observations at standard pressure levels from 925 to 200 hPa during the period from 13 July 2006 to 31 December 2013. It was clear in Fig. 2 that the

temperatures observed by COSMIC agreed quite well with those from RS observations at all standard pressure levels from



925 to 200 hPa, with their correlation coefficients greater than 0.96 at each level. However, the root mean square (RMS) values of the temperature differences between RS and COSMIC observations were non-uniform from 925 to 200 hPa. The RMS decreased from 2.04 °C at 925 hPa to 1.51 °C at 400 hPa, while it increased to 1.74 °C at 200 hPa. Therefore, large discrepancy between RS and COSMIC derived temperature was observed at the lower and upper levels, and a minimum

RMS of 1.51 °C was detected at 400 hPa. The above temperature differences at the lower levels may be explained by the significant systematic negative bias (N-bias) remains in derived refractivity profiles in the atmospheric boundary layer (ABL) (e.g., Xie et al., 2010). Moreover, there were significant representativeness errors between the two types of soundings since the RS is a series of point measurements that drift as they ascend through the atmosphere, whereas COSMIC actually measures the averages over finite volumes (on the order of 300 km) of the atmosphere (Kuo et al., 2004; Anthes et al., 2011).

*[Insert Fig. 2 here]*

**3 Arctic seasonal mean temperature and anomaly profiles from RS and COSMIC observations in 2007–2012**

Using the COSMIC data as reference, previous studies (e.g., Sun et al., 2010; 2013) showed that the RS radiation-induced biases differ between RS type and vary with time. An effective way to estimate the mean temperature variations over the Arctic was to remove the inhomogeneities in RS data. In this Section, the seasonal mean temperatures and anomalies from

COSMIC and homogenized RS data were investigated, and used to compare the abilities of COSMIC and homogenized RS data in revealing Arctic temperature changes. As described in Section II, only an average of about 40 spatio-temporally synchronized matchups between RS and COSMIC temperature profiles at 925 hPa were obtained at a single RS site per day. The abilities of COSMIC observations in characterizing the Arctic temperature variations may need to be further investigated, since the COSMIC observations were more widely distributed than RS observations in both the temporal and

spatial domains. Unlike the temperature comparisons made in Section II, where the measurements were matched both temporally and spatially, another two schemes were designed in this Section. It was important to note that the homogenized RS data used in the rest of the paper were generated with updated Radiosonde Observation Correction Using Reanalyses (RAOBCORE; Haimberger et al., 2007) and Radiosonde Innovation Composite Homogenization (RICH) software packages (Haimberger et al., 2012) to remove the inhomogeneity errors due to the irregular distribution of RS stations and constant

changes of instruments in space and time. The homogenized RS products from RAOBCORE and RICH software packages were generated with global radiosonde temperature dataset back to 1958, which have been verified with temperature datasets based on Advanced Microwave Sounding Unit (AMSU) radiances (Haimberger et al., 2012).

1) Scheme I: The seasonal mean temperatures and anomalies during 2007–2012 over the study area were first estimated from all available homogenized RS observations, which were further compared with those derived by spatially synchronized

COSMIC observations. In this Scheme, a time window of 2 h was eliminated during the matching process between RS and COSMIC observations, and all COSMIC observations in a circle of radius 100 km at each RS site were used.



2) Scheme II: The seasonal mean temperatures and anomalies from homogenized RS observations in Scheme II were obtained as in Scheme I. However, the seasonal mean temperatures and anomalies from COSMIC observations were generated at 5 × 5 degree grids for each standard level at 925–200 hPa in 2007–2012. In this Scheme, the original temporal (i.e., a time window of 2 h) and spatial (i.e., a circle of radius 100 km) limitations during the matching process in Scheme I are both extended, and the gridded COSMIC observations were compared with the spatially located RS data.

### 3.1 Seasonal mean temperature profiles from 2007 to 2012 in the Arctic

In Fig. 3, the seasonal mean temperature profile differences between spatially synchronized RS and COSMIC observations (i.e., Scheme I) at 925–200 hPa in 2007–2012 over the Arctic were shown. It should be noted that three-month seasons were defined as March–May (spring), June–August (summer), September–November (autumn) and December to the following February in the next year (winter). It was clear in Fig. 3 that the seasonal mean temperature differences were still remarkable at the low levels after increasing the number of COSMIC observations in the time domain, especially for the bottom level of 925 hPa. The mean temperature differences were less than about ±1.0 °C at 850–200 hPa at all seasons except the autumn seasons of 2010 and 2011. The RMS and mean difference (MD) of the seasonal temperature and anomaly at 925–200 hPa between RS and COSMIC observations in Schemes I in 2007–2012 were listed in Table I. As we can see in Table II, the RMS values in Schemes I were less than 0.6 °C at all levels except for a RMS of 1.32 °C observed at 925 hPa. The best agreement was achieved at 300 hPa with a RMS of 0.32 °C, followed by 250 and 400 hPa (i.e., a RMS of 0.40 and 0.42 °C, respectively). In addition, the MDs between RS and COSMIC observations in Schemes I were within ±0.14 °C at 925–200 hPa except for a MD of -0.32 °C detected at 700 hPa.

*[Insert Fig. 3 here]*

*[Insert Table I here]*

Considering that the RO events are globally quasi-random distributed, the number of COSMIC observations matched around the RS sites is still rather low. In order to assimilate more COSMIC observations during the comparison, the COSMIC data were collected at 5 × 5 degree grids at the ROI for further comparison. The comparison of seasonal mean temperature profiles from RS and COSMIC observations (i.e., Scheme II) was illustrated in Fig. 4. The seasonal mean temperatures from COSMIC observations at 5 × 5 degree grids appeared to be systemically colder than those from RS observations in 2007–2012, which was especially obvious at the bottom levels of 925–700 hPa in all seasons except winters. The seasonal mean temperature differences became weaker for the levels of 600–200 hPa, and a minimum RMS of about 0.35 °C and an absolute minimum MD of 0.03 °C were both observed at 250 hPa (see Table I), respectively.

Comparison of the results of seasonal mean temperature profiles at 925–200 hPa from RS and COSMIC observations in Figs. 3–4 suggested that larger RMS and MD (see also Table I) were detected at almost all levels in Scheme II compared to Scheme I. The discrepancy in Scheme II was understandable because the two observations were not spatially synchronized,





i.e., the average temperature from COSMIC data was taken over an area of 5 × 5 degrees rather than over the locations of RS sites. Therefore, it is suggested that the seasonal mean temperature deviations between RS and COSMIC observations in Scheme II cannot be relieved even after more COSMIC observations were used.

*[Insert Fig. 4 here]*

**3.2 Seasonal temperature anomalies from 2007 to 2012 in the Arctic**

In this Subsection, the seasonal temperature anomaly profiles at 925–200 hPa from RS and COSMIC data were compared. The anomalies were calculated as the departure from an average. For example, the temperature anomaly for summer 2007 was obtained by subtracting the mean temperature profile of the summer months of 2008–2012 from the mean temperature profile calculated for summer 2007. In Fig. 5, the Arctic seasonal temperature anomaly differences between spatially

synchronized RS and COSMIC observations (i.e., Scheme I) at 925–200 hPa of 2007–2012 were shown in Fig. 5. The Arctic seasonal temperature anomaly differences from the above two observations showed a large discrepancy at 925 hPa, while consistent results were generally observed with a RMS of less than 0.62 °C for the levels above 925 hPa, and MDs ranged from -0.05–0.15 °C at all levels (see also Table I). However, the anomalies from spatially synchronized COSMIC observations were not stable enough to explain the temperature anomalies at the locations of RS sites, since large variations

were observed at different levels for the seasons of 2007–2012. Taking winter 2011 as an example, a negative anomaly difference of -2.7 °C was found at 925 hPa, while a positive anomaly difference of 1.9 °C was observed at 200 hPa. Therefore, the number of spatially synchronized COSMIC observations within only a circle of radius 100 km may be insufficient to describe the temperature anomalies.

*[Insert Fig. 5 here]*

To evaluate the ability of COSMIC observations for use in temperature anomaly estimation, comparison of Arctic seasonal temperature anomalies was also made (see Fig. 6) according to scheme II. As illustrated in Fig. 6, the temperature anomaly differences were less than about ±1.0 °C at all levels for all seasons from 2007 to 2012, and those anomalies showed better agreement than the results in Fig. 5. Few fluctuations of the seasonal temperature anomaly differences were detected, with the RMS values of seasonal anomaly differences less than 0.43 °C, and MDs within the range of -0.004–

0.08 °C at all levels (see Table I). The RMS values of seasonal anomaly differences were even lower than 0.30 °C at 850–300 hPa, in contrast to a minimum RMS of 0.33 °C estimated in Fig. 5. Furthermore, at the level of 400 hPa, the smallest seasonal temperature anomaly difference from RS and COSMIC observations in Scheme II was observed.

*[Insert Fig. 6 here]*





## 4 Seasonal temperature anomalies at 400 hPa from RS and COSMIC observations during 2007 and 2012

The Arctic warming in recent years has been accompanied by a rapid loss of sea ice, especially during the summer season, which has drawn a lot of attention (e.g., Devasthale et al., 2010; 2013; Kay et al., 2011; Kim et al., 2014; Stroeve et al., 2014). It has been reported using both modeling and observational approaches that, the long-term changes in the Arctic sea

ice and the several SIM events for the 2000s (e.g., the 1st and 2nd lowest Arctic sea ice extents in 2012 and 2007, respectively) were associated with the combined influences of clouds, radiation, circulation, atmospheric preconditioning and ice-albedo feedback, etc. In this Section, we compared the Arctic seasonal temperature anomalies at 400 hPa derived from RS and COSMIC observations during the record minimum sea ice extents of 2007 and 2012, and discussed the performance of RS and COSMIC observations in revealing the Arctic temperature variations during the SIM events.

The seasonal temperature anomalies from COSMIC observations at $5 \times 5$ degree grids (i.e., scheme II) matched well with those from homogenized RS observations at the pressure level of 400 hPa. Therefore, the following analyses of Arctic seasonal temperature anomalies during 2007 and 2012 were presented at 400 hPa only. It should be noted that our choice was also consistent with previous reports that the Arctic warming signal was observed from the surface up to 400 hPa (Devasthale et al., 2010; 2013). The spatial distributions of the Arctic seasonal temperature anomalies at 400 hPa in 2007

from RS and COSMIC observations as well as their differences were shown in Fig. 7. The derived anomalies from the two observations showed similar distributions at all RS sites for the seasons in 2007. Despite a maximum negative and positive anomaly difference of -2.76 and 2.97 °C observed in winter 2007, the overall temperature anomaly difference was about only 0.11, 0.08, 0.16 and 0.23 °C in all seasons, respectively. While the RS observations can describe the temperature anomalies over the land areas only, the temperature anomaly distributions derived from COSMIC observations at 5 degree

spatial resolution revealed increased details about the state of Arctic atmosphere. Taking the temperature anomalies in summer 2007 as example, conspicuous positive anomalies were detected from COSMIC observations over the East Siberian Sea and Beaufort Sea, which was consistent with the analysis from AIRS data (Devasthale et al., 2010) and may be one of the most important reasons for the SIM of summer 2007. However, no prominent warming signal was observed by coastal RS sites, and only moderate positive anomalies were found at Barrow, USA (71.3 °N, -156.8 °W) and Cherskiy, Russia

(68.8 °N, 161.3 °E). Therefore, the spatially scattered RS observations over the land may fail to depict the details of Arctic temperature variations.

*[Insert Fig. 7 here]*

In Fig. 8, comparison of seasonal temperature anomalies in 2012 at 400 hPa from RS and COSMIC observations were illustrated. The anomalies from those observations showed good consistency in 2012, with the overall mean anomaly

difference of only 0.32, -0.11, 0.08 and -0.10 °C in each season, respectively. Moreover, the positive anomalies appeared to be wider in spring 2012 (see Fig. 8(a)) than in spring 2007 (see Fig. 7(a)), and the negative anomalies appeared to be weaker in spring 2012 than in spring 2007, which were consistent with the analysis from AIRS data in Devasthale et al. (2013) and





could result in the advanced sea ice melt in spring 2012. The anomalies from RS observations shown in Fig. 8(b) revealed positive signal in Greenland, Longyearbyen and Franz Josef Land, they however provided little information about the temperature variations over the ocean areas. During the autumn of 2012, the warming signal from COSMIC observations were detected over oceans such as Baffin Bay, Chukchi Sea, East Siberian Sea, Laptev Sea and Kara Sea (see Fig. 8(g)),

while the positive anomalies from RS observations were only observed over land areas including Alaska, Far East of Russia, East Siberia and southern Greenland (see Fig. 8(h)). Although it was found from both RS and COSMIC observations that the positive anomalies in autumn were wider and stronger than those in summer, which played an important role in preventing sea ice build-up in autumn 2012 as reported from AIRS data in Devasthale et al. (2013), less guidance from RS observations was obtained due to their poor spatial resolution. Therefore, despite the anomalies from COSMIC and homogenized RS data

presenting similar patterns over the land area, the wider coverage of COSMIC observations showed advantages of revealing the temperature variations over both land and ocean areas, which could be helpful to understand more details about Arctic climate change.

    *[Insert Fig. 8 here]*

**5 Conclusions**

The Arctic air temperature variations play an important role in Arctic climate change and related processes. In this paper, comparisons of Arctic air temperature profiles at 925–200 hPa from RS and COSMIC observations have been conducted. The accuracy of RS and COSMIC derived air temperature profiles were evaluated via comparisons with temporally and spatially synchronized GRUAN observations. Additionally, comparisons of temperature profiles from COSMIC and different types of RS were made to understand the performance of different RS types. Moreover, the Arctic seasonal mean

temperature and anomaly differences from COSMIC and homogenized RS observations were analyzed to understand the ability of the two observations in monitoring changes in Arctic air temperature. Furthermore, the Arctic seasonal temperature anomalies from COSMIC and homogenized RS observations during 2007 and 2012 were compared to investigate the ability of COSMIC observations in revealing the temperature variations during the SIM events. Our findings from this study can be summarized as follows:

(1) Comparison of RS and COSMIC detected temperature profiles at 925–200 hPa with synchronized GRUAN observations showed that the STDs of RS detected temperature profiles were less than 0.3 °C at 850–200 hPa, and the STDs of COSMIC derived "wet" temperature profiles varied from 0.66 to 2.4 °C. The larger STD differences from COSMIC and GRUAN may result from their spatial and temporal mismatch and systemic errors of COSMIC observations. In addition, large discrepancies between RS and COSMIC derived temperature were observed at the lower and upper levels, with a

minimum RMS of 1.51 °C was detected at 400 hPa.





(2) Comparison of seasonal mean temperature and anomaly from COSMIC and homogenized RS observations in Scheme I showed that despite the mean temperature and anomaly differences from the two observations being generally less than ±1.0 °C at 850–200 hPa for all the seasons in 2007–2012, large variations of those differences were observed at different levels. In contrast, the mean temperature and anomaly difference comparisons exhibited more stable performance at different

levels in Scheme II than in Scheme I. Comparison of seasonal mean temperatures in Scheme II suggested that the spatially synchronized COSMIC observations around RS sites may be insufficient to describe the small-scale spatial structure of temperature variations. Moreover, comparisons of seasonal temperature anomalies in Scheme II indicate that the 5 × 5 degree gridded COSMIC observations were able to provide stably estimates of the seasonal Arctic temperature anomalies, with a RMS of less than 0.43 °C at 925–200 hPa compared with the results from homogenized RS observations.

(3) Comparison of the Arctic seasonal temperature anomaly distributions at 400 hPa from COSMIC and homogenized RS observations in Scheme I during the SIM of 2007 and 2012 showed that the COSMIC derived temperature anomalies can provide more details about Arctic temperature variations. As such, the monitoring of atmospheric preconditioning with RO observations could be a complementary source of information in understanding the Arctic upper-air temperature variations and related climate change.

It is worth mentioning that despite the sparsely distribution of COSMIC profiles over the inner Arctic region, incorporation of several other RO missions such as SAC-C (Scientific Application Satellite-C), CHAllenging Mini-satellite Payload (CHAMP), Gravity Recovery and Climate Experiment (GRACE), Meteorological Operational satellite program (MetOp)-A/B, TerraSAR-X and Korea Multi-Purpose Satellite-5 (KOMPSAT-5) could be helpful to relieve this shortcoming. In addition, the increasing number of future RO missions may be also helpful (e.g., COSMIC II, Climate Community

Initiative for Continuing Earth Radio Occultation (CICERO), Meteorological Operational satellite programme (MetOp)-C and Spain's PAZ). Furthermore, with the development of other GNSS (e.g., Russia's GLONASS, Europe's GALILEO and Chinese Beidou), more RO observations will be available, which will be more beneficial to the understanding of Arctic climate change.

It should also be noted that despite the validation in this study of the performance of RO derived temperatures at 400 hPa

in revealing the thermodynamic state of the Arctic atmosphere via comparisons with those from RS observations, the RO observations fail to provide sufficient credible clues at the lowermost troposphere due to the N-bias in the ABL. The state of the Arctic surface atmosphere, however, is one of the most important issues to understand the interactions between the atmosphere and sea-ice, and thus the Arctic climate feedbacks. Therefore, other information is recommended to be accompanied with RO observations to further study the Arctic climate change, which would be an important issue in the

future.

*Acknowledgements.* This work was funded by the Global Change Research Program of China (2015CB953900), National Natural Science Foundation of China (Nos. 41506211 and 41276197), Shanghai Oriental Scholar Program (No. 2012-58), Shanghai Sailing Program (No. 14YF1410200), Innovation Program of Shanghai Municipal Education Commission (Nos. 14YZ118 and 14ZZ148). We thank Leopold





Haimberger (University of Vienna) for providing us with the RAOBCORE and RICH adjusted RS data and Jason M. Boucher (Northeast Fisheries Science Center, National Oceanic and Atmospheric Administration) for his advices on English grammar and expressions. We are grateful to the COSMIC Data Analysis and Archive Center (CDAAC) and Integrated Global Radiosonde Archive (IGRA) for providing their data.

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

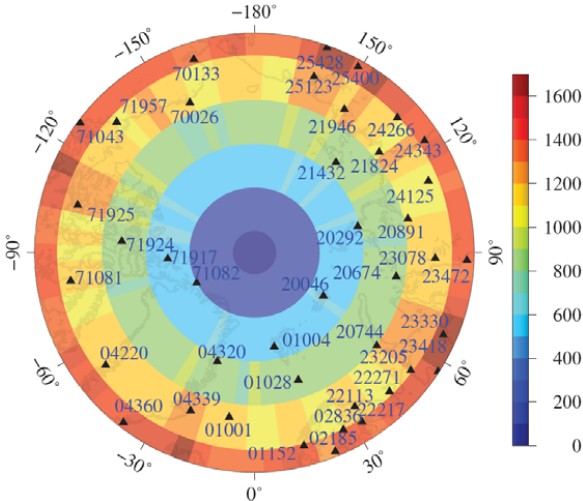

10   **Figure 1: Spatial distribution of the number of COSMIC observations superimposed on coastlines of the Arctic region from July 13, 2006 to December 31, 2013. The blue numbers are the World Meteorological Organization (WMO) identifications of RS sites,and black solid triangles denote the RS sites.**





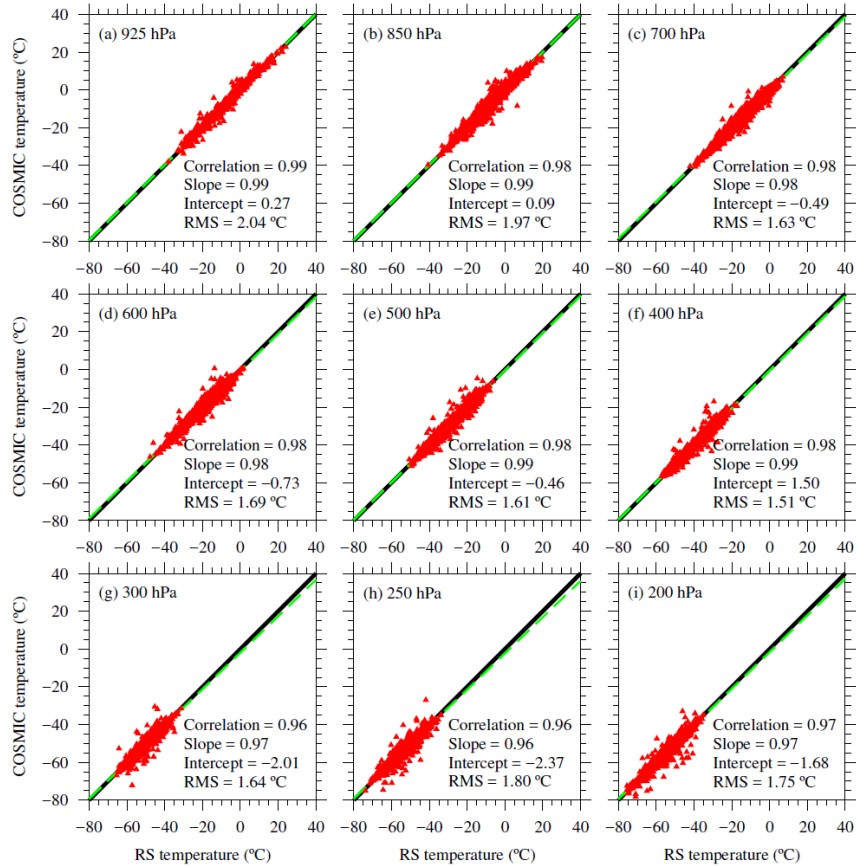

**Figure 2: Scatter plots of Arctic mean temperature profiles between RS and COSMIC observations at different pressure levels from 925 to 200 hPa. The linear regression is shown as the green dashed line, and the black line is the zero bias.**





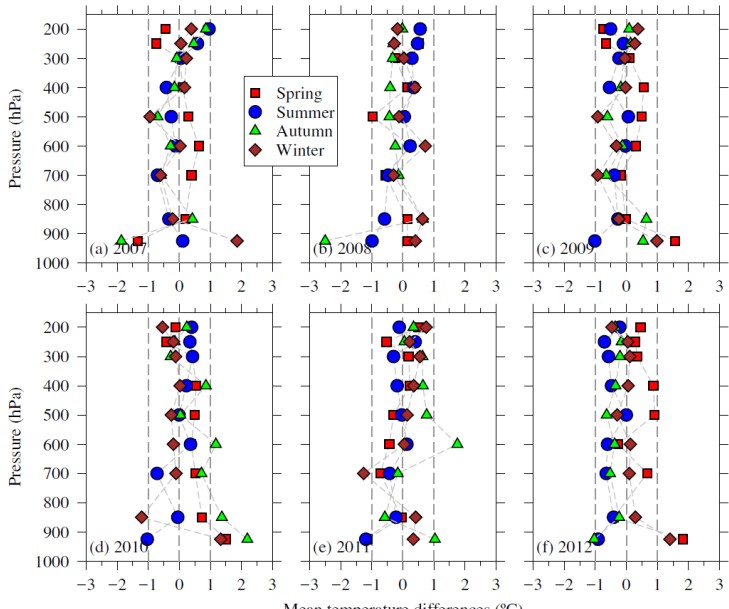

**Figure 3: The differences of Arctic mean temperature profiles at 925–200 hPa between RS and COSMIC observations in Scheme I in each season from 2007 to 2012. Red solid squares, blue solid circles, green solid triangles and brown solid diamonds are the discrepancies in spring, summer, autumn and winter, respectively.**





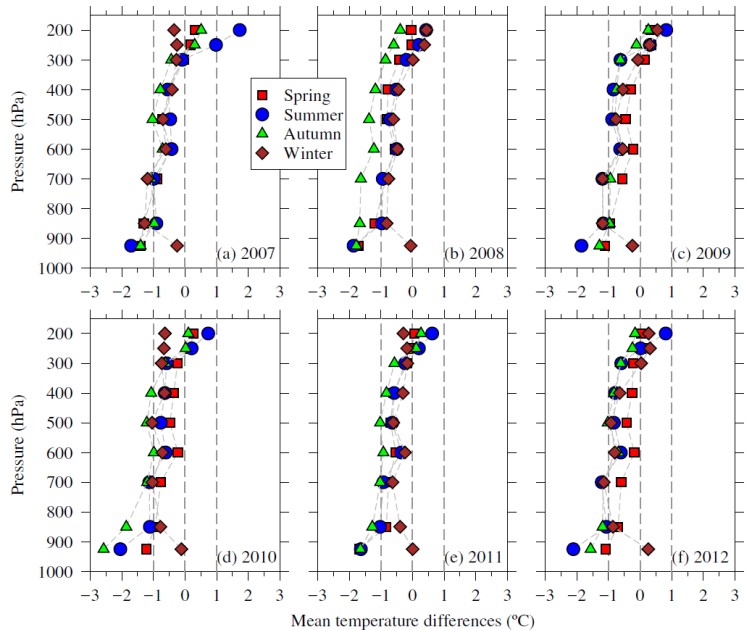

Figure 4: Same as in Figure 3, but for the differences between RS and COSMIC observations in Scheme II.





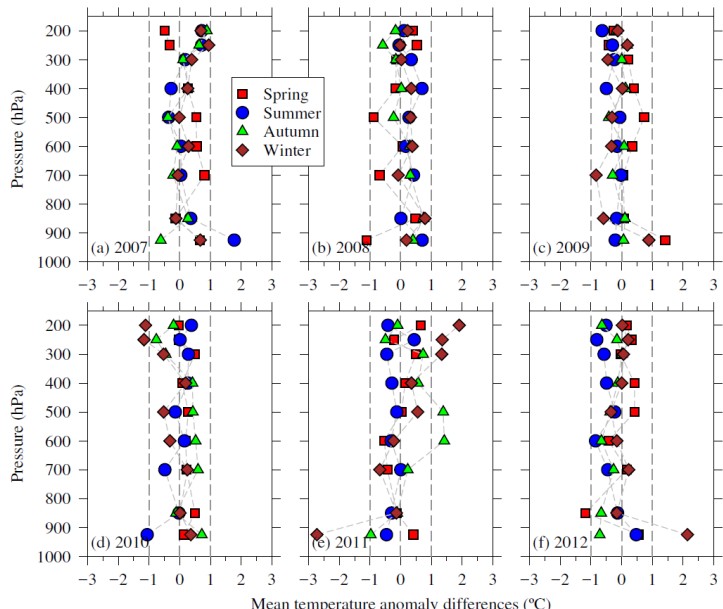

**Figure 5: Same as in Figure 3, but for the differences of Arctic temperature anomalies.**





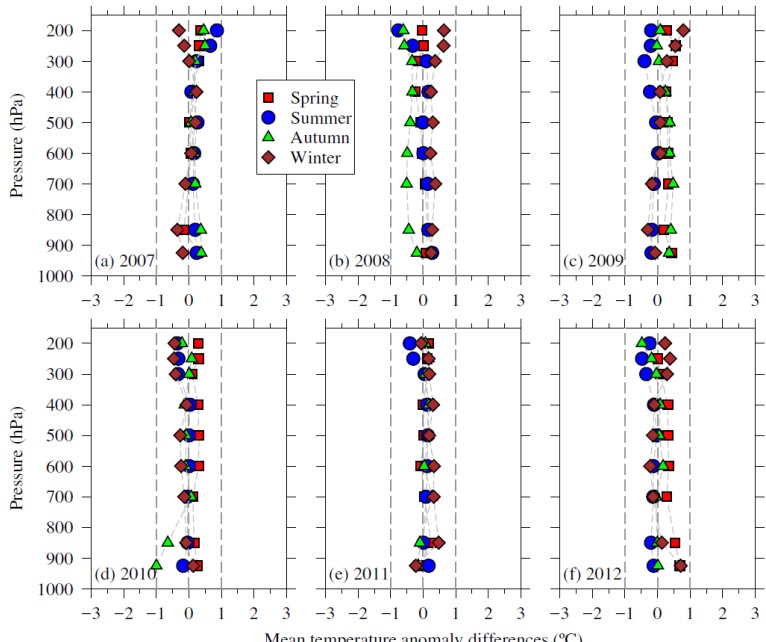

**Figure 6: Same as in Figure 4, but for the differences of Arctic temperature anomalies.**





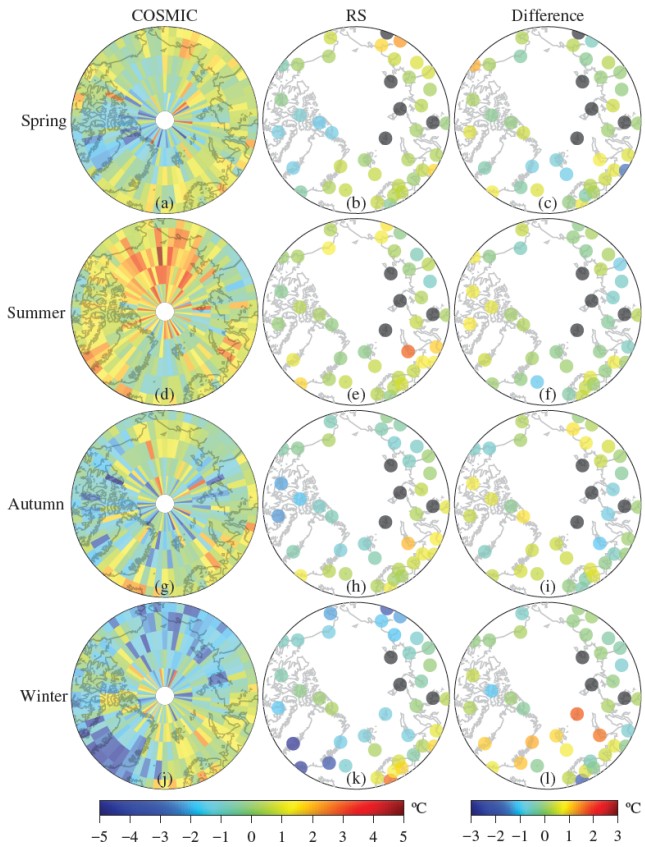

**Figure 7:** The horizontal distributions of seasonal temperature anomalies superimposed on Arctic coastlines from COSMIC (left column) and RS (middle column) observations in Scheme II, as well as their differences (right column) at 400 hPa in 2007. Note that the black dots in the middle and right columns are due to missing data at the RS sites.





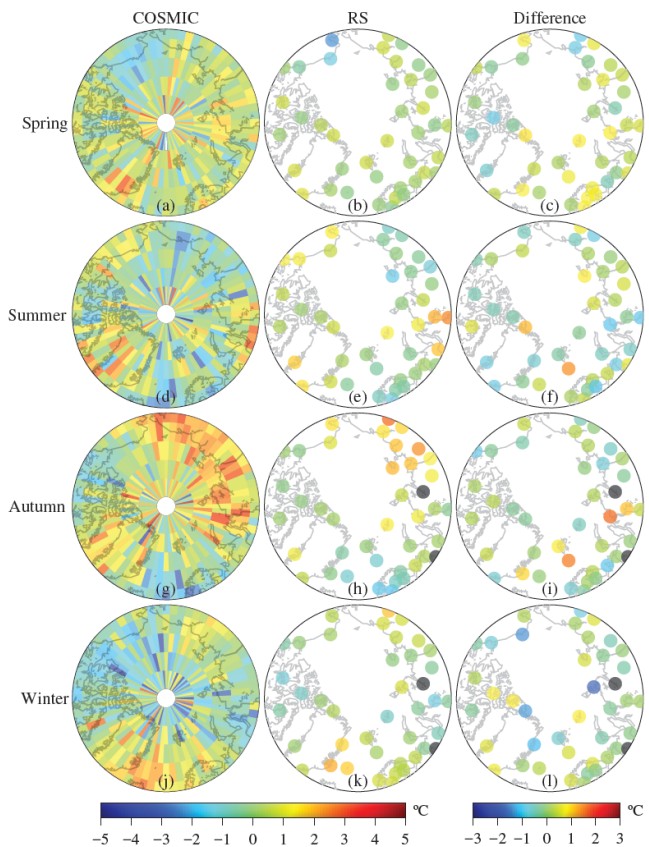

Figure 8: Same as in Figure 7, but for the year of 2012.



**Table 1: The RMS and mean difference (MD) of the seasonal mean temperature and anomaly differences at 925–200 hPa between RS and COSMIC observations in 2007–2012. (unit: °c)**

| Pressure level (hPa) | Temperatures | | | | Temperature anomalies | | | |
|---|---|---|---|---|---|---|---|---|
| | Scheme I (Fig. 3) | | Scheme II (Fig. 4) | | Scheme I (Fig. 5) | | Scheme II (Fig. 6) | |
| | RMS | MD | RMS | MD | RMS | MD | RMS | MD |
| 925 | 1.32 | 0.09 | 1.47 | -1.25 | 1.03 | 0.15 | 0.36 | 0.08 |
| 850 | 0.54 | 0.03 | 1.09 | -1.05 | 0.42 | -0.03 | 0.30 | 0.03 |
| 700 | 0.59 | -0.32 | 1.02 | -0.99 | 0.41 | -0.05 | 0.23 | 0.06 |
| 600 | 0.54 | 0.13 | 0.72 | -0.66 | 0.46 | 0.02 | 0.21 | 0.06 |
| 500 | 0.52 | -0.14 | 0.84 | -0.81 | 0.49 | 0.02 | 0.20 | 0.07 |
| 400 | 0.42 | 0.12 | 0.71 | -0.67 | **0.33** | 0.13 | **0.19** | 0.07 |
| 300 | **0.30** | 0.04 | 0.48 | -0.40 | 0.44 | 0.08 | 0.25 | 0.04 |
| 250 | 0.40 | **-0.02** | **0.35** | **0.03** | 0.58 | **0.01** | 0.37 | 0.06 |
| 200 | 0.49 | 0.12 | 0.56 | 0.24 | 0.62 | 0.05 | 0.43 | **-0.004** |