# Peer review of "Comparison of the Arctic upper-air temperatures from radiosonde and radio occultation observations"

_Atmospheric Measurement Techniques, 2016_

## Referee Comment (RC1) · Y. Liu (Referee) · 29 Aug 2016

Y. Liu (Referee)

yinghuil@ssec.wisc.edu

This manuscript makes an effort to apply radio occultation (RO) temperature profiles in revealing the Arctic temperature variations in the troposphere associated with the sea ice change. For this purpose, the authors compare the performance of RO temperature profiles and radiosonde (RS) temperature profiles in the troposphere from 900 t0 250 hPa. The way to compare RO and RS profile performance using two schemes is plausible. The conclusion of the added value of RO in studying Arctic temperature variation is reasonable.

My specific comments are:

1. line 20 on page 1, this sentence is confusing by just reading the abstract, the authors need revise this by pointing out what details the RO can provide. 2. Line 29 on page 1,

it is arguable that radiosonde has poor temporal resolution. 3. Line 21 on page 2, this sentence is confusing, need revision. 4. Line 25 on page 2, after "furthermore" needs revision 5. line 7 on page 4, same as 2 6. line 30 on page 4, I suggest the authors delete sentences as "...agree well...", instead, just providing the detailed statistics here and other places in this manuscript 7. Line 12 on page 5, I found this sentence is not clear. 8. Line 16 on page 5, I do not see where the authors describe this in section II. 9. Line 19 on page 5, this sentence is not true IMO. 10. Line 24 on page 5, what is RAOBCORE and RICH? 11. Line 14 page 6, table I. 12. Line 23 on page 6, what is ROI? 13 Line 2 on page 7, you might want rewrite this sentence. 14. Line 5-10 on page 7, can you define how the anomalies are calculated? 15 Line 10 -25, I suggest the authors add a table to show the values of the anomalies at those RS stations and correspondent grid. Also, provide statistical details. 16, Line 23 on page 8, should be autumn 2007. 17, line 29 on page 8, same as 16. 18 Line 18-19 one page 9, this sentence is confusing. 19 Line 10 on page 10, this sentence is confusing 20 Line 27 on page 10, considering replacing "surface atmosphere" 21 Figure 7 and 8, the black dots appear in grey color.

---

## Referee Comment (RC2) · Anonymous Referee #2 · 29 Aug 2016

The authors make great efforts to compare the arctic upper-air temperature from the radiosonde and cosmic RO wetPrf data. Some results have been shown. But many similar works have been published. So I think the authors should add some novel study in this Paper.

———————

---

## Author Comment (AC1) · 27 Sep 2016

This manuscript makes an effort to apply radio occultation (RO) temperature profiles in revealing the Arctic temperature variations in the troposphere associated with the sea ice change. For this purpose, the authors compare the performance of RO temperature profiles and radiosonde (RS) temperature profiles in the troposphere from 900 to 250 hPa. The way to compare RO and RS profile performance using two schemes is plausible. The conclusion of the added value of RO in studying Arctic temperature variation is reasonable. Answer: Thank you very much for your positive comments on our manuscript. We have further improved the manuscript following your suggestions. My specific comments are: 1. Line 20 on page 1, this sentence is confusing by just reading the abstract, the authors need revise this by pointing out what details the RO can provide. Answer: We appreciate the comment. According to the Reviewer's

suggestion, we have explained the details obtained from the RO observations in the revised manuscript, i.e., 'Results revealed that similar Arctic temperature variation patterns can be obtained from both RS and COSMIC observations over the land area, while extra information can be further provided from the widely covered COSMIC observations.' 2. Line 29 on page 1, it is arguable that radiosonde has poor temporal resolution. Answer: Thanks very much for your rigorous thinking. In the revision, the description of temporal resolution was deleted. 3. Line 21 on page 2, this sentence is confusing, need revision. Answer: We have rewritten this sentence. 4. Line 25 on page 2, after "furthermore" needs revision. Answer: We are sorry for the carelessness in the manuscript. The comparisons of temperature measurements from COSMIC and different types of RS were not made in the original manuscript, and we therefore delete this sentence in the revised version. 5. Line 7 on page 4, same as 2. Answer: We have revised the description. 6. Line 30 on page 4, I suggest the authors delete sentences as "...agree well...", instead, just providing the detailed statistics here and other places in this manuscript. Answer: Thanks for your comment. A detailed description has been provided in the revised manuscript. 7. Line 12 on page 5, I found this sentence is not clear. Answer: We have reworded the sentence in the revision, i.e., 'The value of operational RS observations for climate monitoring were strongly hindered by numerous and poorly documented changes in instrumentation and operational procedures (Titchner et al., 2009). In addition, differences between radiosondes from different manufacturers complicated the comparison of data records from different sources (Moradi et al., 2013). Therefore, it may be arbitrary to use the RS records directly for long-term climate monitoring and trend detection.' 8. Line 16 on page 5, I do not see where the authors describe this in section II. Answer: We are sorry for the carelessness. We have rechecked the programs and found that the number of spatio-temporally synchronized RS and COSMIC temperature measurements at 925 hPa was 547 (see Table R1), and therefore an average of about 13 observations (547/41 = 13.341) were obtained per single RS site during the period from 13 July 2006 to 31 December 2013. In the revised manuscript, we have corrected the descriptions. 9. Line 19 on page 5,

this sentence is not true IMO. Answer: The description of RS observations in temporal domain has been deleted in the revised version. 10. Line 24 on page 5, what is RAOB-CORE and RICH? Answer: One of the major challenges for using RS records as a reference may be their the lack of absolute accuracy, since radiosondes suffered suffer from radiation errors in temperature measurements and had have various errors/biases in humidity data, especially in the upper troposphere (e.g., Wang et al., 2003). Therefore, the RAOBCORE and RICH software packages are incorporated in this study to errors/biases in RS temperature measurements for further analysis. 11. Line 14 page 6, table I. Answer: We have corrected the 'Table II' to 'Table I' in the revision. 12. Line 23 on page 6, what is ROI? Answer: The full name of the abbreviation 'ROI' has been added in the revised manuscript. 13. Line 2 on page 7, you might want rewrite this sentence. Answer: We have rewritten the sentence in the revision. 14. Line 5-10 on page 7, can you define how the anomalies are calculated? Answer: The temperature anomaly is the difference between the long-term average temperature (sometimes called a reference value) and the temperature that is actually occurring. In other words, the long-term average temperature is one that would be expected; the anomaly is the difference between what you would expect and what is happening. In the revision, the calculation of temperature anomaly has been clarified. 15. Line 10-25, I suggest the authors add a table to show the values of the anomalies at those RS stations and correspondent grid. Also, provide statistical details. Answer: Thanks very much for your suggestion. The temperature anomalies at RS and correspondent RO grid on 2007 (see Table R2) and 2012 (see Table R3) were illustrated. However, Tables R2 and R3 were not incorporated in the revised manuscript because it may be redundant to list all values at each RS site. Furthermore, the temperature anomaly differences for each site have been illustrated in the right columns in Figs 7 and 8. 16. Line 23 on page 8, should be autumn 2007. 17. Line 29 on page 8, same as 16. Answer: Revisions have been made according to above two comments. 18. Line 18-19 one page 9, this sentence is confusing. Answer: We are sorry for the carelessness. The comparisons of temperature measurements from COSMIC and different types of RS were not made

in the original manuscript, and we therefore delete this sentence in the revised version. 19. Line 10 on page 10, this sentence is confusing. Answer: We have reorganized this sentence in the revision. 20. Line 27 on page 10, considering replacing "surface atmosphere" Answer: We have corrected in the revision. 21. Figure 7 and 8, the black dots appear in grey color. Answer: We have corrected the 'black' to 'grey' in the revision.

Reference Wang, J. H., Carlson, D. J., Parsons, D. B., Hock, T. F., Lauritsen, D., Cole, H. L., Beierle, K., and Chamberlain, E.: Performance of operational radiosonde humidity sensors in direct comparison with a chilled mirror dew-point hygrometer and its climate implication, Geophys Res Lett, 30, 2003.

Please also note the supplement to this comment:
http://www.atmos-meas-tech-discuss.net/amt-2016-232/amt-2016-232-AC1-supplement.zip

---

## Author Comment (AC2) · 27 Sep 2016

The authors make great efforts to compare the arctic upper-air temperature from the radiosonde and cosmic RO wetPrf data. Some results have been shown. But many similar works have been published. So I think the authors should add some novel study in this Paper. Answer: Thanks very much for your encouragements on our manuscript. We admit that many works about comparing the temperature measurements between radiosonde (RS) and COSMIC observations have been done. However, the abilities of COSMIC observation on revealing the Arctic climate change may need to be further investigated. Despite the incapability of radio occultation (RO) observations on estimating the surface atmospheric temperature has been demonstrated from previous studies, we found in the manuscript that the RO derived seasonal temperature anomalies at 5 × 5 degree grids from 2007 to 2012 at 400 hPa

show the best agreement with RS results, with a RMS and mean difference of 0.19 and 0.07 °C, respectively. Furthermore, analysis of seasonal temperature anomalies from 5 × 5 degree gridded COSMIC observations at 400 hPa during the Arctic sea ice minimum (SIM) of 2007 and 2012 shows that the RO observations could be helpful to reveal the Arctic sea ice decline in 2007 and 2012 on atmospheric thermodynamics. However, the spatially scattered RS observations over the land fail to depict the details of Arctic temperature variations and therefore the Arctic sea ice change. As such, it can be found in the manuscript that the wider coverage of COSMIC observations showed advantages of revealing the temperature variations over both land and ocean areas, which could be helpful to understand more details about Arctic climate change.

Please also note the supplement to this comment:
http://www.atmos-meas-tech-discuss.net/amt-2016-232/amt-2016-232-AC2-supplement.zip

---

## Referee Comment (RC3) · Anonymous Referee #3 · 11 Oct 2016

**Review of "Comparisons of Arctic upper-air temperatures from radiosonde and radio occultation observations" by Chang et al., submitted to AMTD**

**General comments:**

I have already served as a reviewer of this paper for a different journal where I, during three rounds of review pointed out various, in parts major issues with the paper.
Unfortunately the version of the paper published here does not include some of the major comments I already provided for the other journal. This makes me question if the authors actually value the hours of time reviewers are spending to provide feedback on their work. Here I am providing another review of this paper, but I marked all the parts I could copy from my last review in red.

Unfortunately this paper is not very well written and in parts hard to read. In addition some sentences can be misunderstood, which is very undesirable for a scientific paper.

I also realized that some sections are missing/deleted. The introduction states that section 2 will describe a comparison with the GCOS Reference Upper-Air Network, but this comparison is not in the paper. I do know that this comparison was done in earlier versions of the paper, but the authors seem to have taken it out without adapting the introduction and conclusion. I don't think enough care is taken in during the preparation of this paper.

After recognising that Fig.1 has issues pointed out in the specific comments I stopped reviewing this paper. Although I had pointed out to the authors that something is wrong with Fig.1 during a review of the same paper in a different journal, the issues are still not fixed. I have lost the confidence that the authors provide a carefully investigated study here.
While the final decision should be up to the editor, I would advise to reject this paper due to a wrong figure, poor writing and because this paper is not providing outstanding new findings.

**Specific comments:**
The authors point out many times that they aim to understand the performance of both radiosonde and radio occultation measurements/retrievals in in describing the Arctic upper-air temperature. I don't understand how this is possible. How can you make a statement about the performance of two instruments when comparing them given that non of them is used as a reference.

Another point is, that in general, lower atmospheric RO temperature (wet temperature) profiles contain a priori information from a model or from a climatology like ERA Interim. In order to get a temperature in lower Arctic levels, an estimate of the Arctic humidity will be used in the retrieval and this humidity estimate will be not always be correct (and depend on the reanalysis/model used). I am not sure how much is really gained when using the RO wet temperatures compared to using the reanalysis itself. Also the radiosondes are included in this reanalysis, thus being included in the a priori for the RO retrieval.

When having a close look at the figure 1, I realised that the positions of the GRUAN and radiosonde stations do not agree with the 5 by 5 degree grid that you describe about in the text.

- Figure 1: Is the blue dot in the middle the pole? Is that 90degrees? In figure 9 you make this inner circle white as it is only the pole, so why not doing this here?
- But anyway there is a bigger issue with this figure. There should be 5degree by 5degree grids. So the most inner circle (after the blue dot that I think should be the pole) would go from 90-85, the second from 85-80, then third 80-75, forth 75-70 and fifth 70-65. But comparing this expectation with the position of three sites shows me that something is not

correct. The bins in the following description are counted from inside (but the most inner circle is ignored as I think its the pole, if its not the pole, this would not explain the positions either.)

- Sodankyla (WMO station ID 02836) 67.37° , 26.63° at the middle of the fifth bin (I am not sure which of the two possible dots should be Sodankyla, but both of them are at around 70N)
- Ny-Ålesund (01004) 78.92°, 11.92° should be at the inner end of the third bin. This is not the case. I don't know why this is not the case, but e.g. NYA actually appears in the second bin.
- Barrow (70026) 71.28°, -156.78° should be at the outer end of the forth bin, but is at the beginning of the fourth bin

Unfortunately this major issue was pointed out by myself during the review of this paper for another journal. Still the authors have not fixed it, or clarified (in case I did misunderstood something here). At this stage I decide to finish the review, as I lost the confidence in the carefulness of the authors. I am not sure if the text is simply incorrect or if the figure is wrong, though I have the feeling that in general this work is not done with the needed care to produce high-quality scientific publications.

**technical corrections**

In my opinion this paper is not well written and I will only point this out in parts.
page 1, line 12: "introduced", I would probably write "used" instead
page 1, line 15: "matched" I would probably use the word "agreed" instead
page 1, line 20: "widely covered"? I know you mean that they have a good spatial resolution, but I wouldn't say it like this
page 1, line 27-28: this sentence is not nice. I don't like "traditional tool" and also the radiosonde itself does not have a poor temporal resolution (it measures data every e.g. every second). What you mean is that the RS time series only has one/two profiles every day.
This comment might seem fussy, but this is a scientific paper which is going through peer-review and should be as correct as possible afterwards.
Page 2, line 9: Please rewrite. "The RO techniques **is** based on" ... and also the signal propagates from the GNSS satellite to the LEO satellite (and never the other way around, which I think between implies).
Page 2, line 13-14: This does not only count in the lower troposphere, but at all levels where humidity is not negligible. Please make this clear.
Page 2, line 15-16: This is not correct. You can derive dry temperatures throughout the whole atmosphere, though in the lower levels they will not agree with the physical temperature as they attribute the influence of the water vapour to the temperature.  I would suggest changing the whole paragraph from line 13 onwards to actually clarify what dry and wet temperatures are and where they are valid.
Page 2, line 19: You are saying you compare the RO wet temperatures with the GCOS Reference Upper-Air Network in section 2, but indeed you have cut the section about this comparison out of the paper in this version prepared for AMTD. You mention it again in the conclusion though. I am not sure how that can happen, but it makes me wonder if enough care was taken when preparing this publication.
page 2, line 21: "were used to estimate" are you estimating the temperatures? I thought the CDACC product provides the temperatures? As far as I understand you estimate temperature anomalies.
Page 2, line 24-25: I don't understand how the performance of both measurement systems can be understood. If you are not using any of them as a reference, what can you say about the performance?
Page 14, line 24: "Additionally, the temperature profiles from RS and COSMIC observations were

compared to understand their performance in describing the Arctic atmospheric temperature." again I don't think you compare the performance, how do you know about the performance? what is your reference? I think you actually compare the the COSMIC with the RS observations.

Page 3, line 4-6: Too long sentence, break up in two.

Page 3, line 8: why only lower troposphere?

Page 3, line 12-13: the abbreviation fr CDACC is wrong. It is the COSMIC Data Analysis and Archive Center

Page 3, line 22-26: This is not essential for the study. Please shorten.

Page 3, line 28-33: I think this is not needed. Just state how many profiles you can use, or just the number per grid might be sufficient.

---

## Author Comment (AC3) · 14 Oct 2016

Please see the attachment.

Please also note the supplement to this comment:
http://www.atmos-meas-tech-discuss.net/amt-2016-232/amt-2016-232-AC3-supplement.pdf

---

## Author Comment (AC4) · 14 Oct 2016

[revised manuscript text omitted]

20   Arctic air temperature.

In this paper, the "wet" temperature profiles from the Constellation Observing System for Meteorology, Ionosphere, and Climate (COSMIC)/Formosa Satellite Mission 3 (hereafter COSMIC) RO observations were used to obtain the Arctic seasonal mean temperatures and anomalies from 2007 to 2012. The COSMIC temperatures and anomalies were then compared with the results from RS observations. This paper was organized as follows. In Section 2, the temperature profiles

25   from RS and COSMIC observations were compared to understand their characteristics in describing the Arctic atmospheric temperature. In order to understand the characteristics of RS and COSMIC observations in monitoring Arctic climate change, Section 3 compared the seasonal mean temperatures and anomalies from spatially matched RS and COSMIC observations, and analyzed their differences between RS and 5 × 5 degree gridded COSMIC observations. Section 4 was devoted to discussing the temperature anomaly differences from RS and 5 × 5 degree gridded COSMIC observations during the record

[revised manuscript text omitted]

The value of operational RS observations for climate monitoring were strongly hindered by numerous and poorly documented changes in instrumentation and operational procedures (Titchner et al., 2009). In addition, differences between

radiosondes from different manufacturers complicated the comparison of data records from different sources (Moradi et al., 2013). Therefore, it may be arbitrary to use the RS records directly for long-term climate monitoring and trend detection. An effective way to estimate the mean temperature variations over the Arctic was to remove the inhomogeneities in RS data. In this Section, the seasonal mean temperatures and anomalies from COSMIC and homogenized RS data were investigated, and

5    used to compare the abilities of COSMIC and homogenized RS data in revealing Arctic temperature changes. Considering only an average of about 13 spatio-temporally synchronized matchups (not shown) between RS and COSMIC temperature profiles at 925 hPa were obtained per single RS site during the period from 13 July 2006 to 31 December 2013, the abilities of COSMIC observations in characterizing the Arctic temperature variations may need to be further investigated, since the COSMIC observations were more widely distributed than RS observations in spatial domains. Unlike the temperature

[revised manuscript text omitted]

Comparison of the results of seasonal mean temperature profiles at 925–200 hPa from RS and COSMIC observations in Figs. 3–4 suggested that larger RMS and MD were detected at almost all levels in Scheme II compared to Scheme I (see also Table I). The discrepancy in Scheme II was understandable because the two observations were not spatially synchronized, i.e., the average temperature from COSMIC data was taken over an area of 5 × 5 degrees rather than over the locations of RS sites. Therefore, it could be concluded that the quasi-random distributed COSMIC observations may be insufficient to describe the small-scale spatial structure of mean temperature variations.

*[Insert Fig. 4 here]*

**3.2 Seasonal temperature anomalies from 2007 to 2012 in the Arctic**

In this Subsection, the seasonal temperature anomaly profiles at 925–200 hPa from RS and COSMIC data were compared. The temperature anomalies are defined by a departure from a reference value or long-term average, and calculated as the difference between the long-term average temperature and the temperature that is actually occurring. 
[revised manuscript text omitted]

---

## Author Comment (AC5) · 14 Oct 2016

General comments: I have already served as a reviewer of this paper for a different journal where I, during three rounds of review pointed out various, in parts major issues with the paper. Unfortunately the version of the paper published here does not include some of the major comments I already provided for the other journal. This makes me question if the authors actually value the hours of time reviewers are spending to provide feedback on their work. Here I am providing another review of this paper, but I marked all the parts I could copy from my last review in red. Unfortunately this paper is not very well written and in parts hard to read. In addition some sentences can be misunderstood, which is very undesirable for a scientific paper. Answer: Thank you very much again for agreeing to review the manuscript after previous three rounds of review. We highly appreciate the comments from the Reviewers, and value the spent

time from the Reviewers on our manuscript. In fact, the previous three rounds of review (especially the comments from the third round) depressed me so much. However, my co-authors and I will never give up and will make our effort to improve the manuscript. Here on behalf of all co-authors, I am continuing to response the comment from the Review, please see details below. I also realized that some sections are missing/deleted. The introduction states that section 2 will describe a comparison with the GCOS Reference Upper-Air Network, but this comparison is not in the paper. I do know that this comparison was done in earlier versions of the paper, but the authors seem to have taken it out without adapting the introduction and conclusion. I don't think enough care is taken in during the preparation of this paper. Answer: Sorry for the carelessness! The comparisons of COSMIC and RS temperature with GCOS observations were deleted in the manuscript. Because only three GCOS sites are available in the Arctic, it may be insufficient to evaluate the performance of COSMIC and RS observations in the Arctic. After recognising that Fig.1 has issues pointed out in the specific comments I stopped reviewing this paper. Although I had pointed out to the authors that something is wrong with Fig.1 during a review of the same paper in a different journal, the issues are still not fixed. I have lost the confidence that the authors provide a carefully investigated study here. Answer: After going over the results in Fig. 1, we realized that the results were illustrated incorrectly due to the mapping software. In the original manuscript, we got the results in Fig.1 with Matlab software first and then generated the Fig. 1 with the Generic Mapping Tools (GMT) software. However, the results from GMT software showed something strange and we do not know how to solve it. Therefore, we regenerate the Figs. 1, 7 and 8 with Matlab software in the revised manuscript, and the shifts of RS position were fixed. While the final decision should be up to the editor, I would advise to reject this paper due to a wrong figure, poor writing and because this paper is not providing outstanding new findings. Answer: We thank the Reviewer for giving the direct suggestion for the Editor. We admit that many works have been done on comparing the temperature measurements between radiosonde (RS) and COSMIC observations on global scale. However, the Reviewer is invited to pay attention to the

explanations of Arctic sea ice extents in 2007 and 2012 with COSMIC observations in the manuscript. As the number of observations in the Arctic is an important issue for the Arctic climate change research, it is demonstrated in this manuscript that the COSMIC data can be a complementary source of information in understanding the Arctic upper-air temperature variations and related climate change. Specific comments: The authors point out many times that they aim to understand the performance of both radiosonde and radio occultation measurements/retrievals in in describing the Arctic upper-air temperature. I don't understand how this is possible. How can you make a statement about the performance of two instruments when comparing them given that non of them is used as a reference. Answer: The eventual purpose of the manuscript is to demonstrate the possibility of COSMIC data in reveal the Arctic climate change, rather than the comparison between COSMIC and RS temperature. In the revised manuscript, the COSMIC and RS temperature measurements were compared, and results show large temperature discrepancies between RS and COSMIC observations were observed at the lower and upper levels, while a minimum RMS of 1.51 °C was detected at 400 hPa. In addition, considering the operational RS observations for climate monitoring were strongly hindered by numerous and poorly documented changes in instrumentation and operational procedures, the homogenized RS data were used for further analysis. Another point is, that in general, lower atmospheric RO temperature (wet temperature) profiles contain a priori information from a model or from a climatology like ERA Interim. In order to get a temperature in lower Arctic levels, an estimate of the Arctic humidity will be used in the retrieval and this humidity estimate will be not always be correct (and depend on the reanalysis/model used). I am not sure how much is really gained when using the RO wet temperatures compared to using the reanalysis itself. Also the radiosondes are included in this reanalysis, thus being included in the a priori for the RO retrieval. Answer: As analyzed in Section 4 of the original manuscript, seasonal temperature anomalies from RS and COSMIC observations at 400 hPa during 2007 and 2012 were used to explain the Arctic sea ice extents in 2007 and 2012. At the layer of 400 hPa, the temperature measurements from reanalysis and RO data are

different from each other. When having a close look at the figure 1, I realised that the positions of the GRUAN and radiosonde stations do not agree with the 5 by 5 degree grid that you describe about in the text. • Figure 1: Is the blue dot in the middle the pole? Is that 90degrees? In figure 9 you make this inner circle white as it is only the pole, so why not doing this here? • But anyway there is a bigger issue with this figure. There should be 5degree by 5degree grids. So the most inner circle (after the blue dot that I think should be the pole) would go from 90-85, the second from 85-80, then third 80-75, forth 75-70 and fifth 70-65. But comparing this expectation with the position of three sites shows me that something is not correct. The bins in the following description are counted from inside (but the most inner circle is ignored as I think its the pole, if its not the pole, this would not explain the positions either.) • Sodankyla (WMO station ID 02836) 67.37° , 26.63° at the middle of the fifth bin (I am not sure which of the two possible dots should be Sodankyla, but both of them are at around 70N) • Ny-Ålesund (01004) 78.92°, 11.92° should be at the inner end of the third bin. This is not the case. I don't know why this is not the case, but e.g. NYA actually appears in the second bin. • Barrow (70026) 71.28°, -156.78° should be at the outer end of the forth bin, but is at the beginning of the fourth bin. Unfortunately this major issue was pointed out by myself during the review of this paper for another journal. Still the authors have not fixed it, or clarified (in case I did misunderstood something here). At this stage I decide to finish the review, as I lost the confidence in the carefulness of the authors. I am not sure if the text is simply incorrect or if the figure is wrong, though I have the feeling that in general this work is not done with the needed care to produce high-quality scientific publications. Answer: As described above, the analysis during the data processing in the manuscript are right. The discouraging performance in Fig. 1 was mainly due to the mapping software. After the results were regenerated with Matlab rather than GMT software in the revision, the above problems were solved. We are sorry for the careless descriptions in the original manuscript. technical corrections In my opinion this paper is not well written and I will only point this out in parts. page 1, line 12: "introduced", I would probably write "used" instead page 1, line 15: "matched"

[Figure]

I would probably use the word "agreed" instead page 1, line 20: "widely covered"? I know you mean that they have a good spatial resolution, but I wouldn't say it like this page 1, line 27-28: this sentence is not nice. I don't like "traditional tool" and also the radiosonde itself does not have a poor temporal resolution (it measures data every e.g. every second). What you mean is that the RS time series only has one/two profiles every day. This comment might seem fussy, but this is a scientific paper which is going through peer-review and should be as correct as possible afterwards. Answer: We appreciate the Reviewer for his/her carefulness, but I think some of the descriptions in the original manuscript is also suitable (e.g., 'introduced'). Despite I am not fully agree with the Reviewer's comments in English written, I revised above 4 comments in the revised manuscript. Page 2, line 9: Please rewrite. "The RO techniques is based on" ... and also the signal propagates from the GNSS satellite to the LEO satellite (and never the other way around, which I think between implies). Answer: We have rewritten the sentence, i.e., "The RO technique was based on the time delays of the radio signal propagating from the GNSS satellite (e.g., Global Positioning System (GPS)) to the receiver placed on a low earth orbit (LEO) satellite (Kursinski et al., 1997). The radio signal was bent by the atmosphere, and the bending angles of the RO signal are derived from the propagation time, which can be precisely measured with atomic clocks." Page 2, line 13-14: This does not only count in the lower troposphere, but at all levels where humidity is not negligible. Please make this clear. Page 2, line 15-16: This is not correct. You can derive dry temperatures throughout the whole atmosphere, though in the lower levels they will not agree with the physical temperature as they attribute the influence of the water vapour to the temperature. I would suggest changing the whole paragraph from line 13 onwards to actually clarify what dry and wet temperatures are and where they are valid. Answer: We have rewritten the sentences. The description on 'lower troposphere' was deleted, and the 'a priori information' was added. Page 2, line 19: You are saying you compare the RO wet temperatures with the GCOS Reference Upper-Air Network in section 2, but indeed you have cut the section about this comparison out of the paper in this version prepared for AMTD. You mention it again

in the conclusion though. I am not sure how that can happen, but it makes me wonder if enough care was taken when preparing this publication. Answer: Sorry for the carelessness! The comparisons of COSMIC and RS temperature with GCOS observations were deleted in the manuscript. Because only three GCOS sites are available in the Arctic, it may be insufficient to evaluate the performance of COSMIC and RS observations in the Arctic. page 2, line 21: "were used to estimate" are you estimating the temperatures? I thought the CDACC product provides the temperatures? As far as I understand you estimate temperature anomalies. Answer: We have corrected the description and similar expressions in the revised manuscript. Page 2, line 24-25: I don't understand how the performance of both measurement systems can be understood. If you are not using any of them as a reference, what can you say about the performance? Page 14, line 24: "Additionally, the temperature profiles from RS and COSMIC observations were I don't think you compare the performance, how do you know about the performance? what is your reference? I think you actually compare the the COSMIC with the RS observations. Answer to the above two comments: In the revised manuscript, we have corrected the expressions. The comparison of COSMIC and RS temperature was implemented in the original manuscript to estimate their agreement, and what we are more concerned about is the possibility of COSMIC data in reveal the Arctic climate change. In the revised manuscript, considering the operational RS observations for climate monitoring were strongly hindered by numerous and poorly documented changes in instrumentation and operational procedures, the homogenized RS data were used for further analysis.

Page 3, line 4-6: Too long sentence, break up in two. Page 3, line 8: why only lower troposphere? Page 3, line 12-13: the abbreviation fr CDACC is wrong. It is the COSMIC Data Analysis and Archive Center Answer to the above three comments: We have made the corrections in the revision. Page 3, line 22-26: This is not essential for the study. Please shorten. Answer: We have deleted the sentence. Page 3, line 28-33: I think this is not needed. Just state how many profiles you can use, or just the number per grid might be sufficient. Answer: We have deleted the sentence.

Please also note the supplement to this comment:
http://www.atmos-meas-tech-discuss.net/amt-2016-232/amt-2016-232-AC5-supplement.zip

―――――――――――――――――